# Mechanistic Study of Triazole Based Aminodiol Derivatives in Leukemic Cells—Crosstalk between Mitochondrial Stress-Involved Apoptosis and Autophagy

**DOI:** 10.3390/ijms21072470

**Published:** 2020-04-02

**Authors:** She-Hung Chan, Wohn-Jenn Leu, Sharada Prasanna Swain, Jui-Ling Hsu, Duen-Ren Hou, Jih-Hwa Guh

**Affiliations:** 1Department of Cosmetic Science, Providence University, 200, Sec. 7, Taiwan Boulevard, Shalu Dist., Taichung 43301, Taiwan; sandychan77@yahoo.com.tw; 2School of Pharmacy, National Taiwan University, No.33, Linsen S. Rd., Zhongzheng Dist., Taipei 100, Taiwan; r00423018@gmail.com (W.-J.L.); d97423004@ntu.edu.tw (J.-L.H.); 3Department of Chemistry, National Central University, No. 300 Jhong-Da Road, Jhong-li, Taoyuan 32001, Taiwan; nobelsarada2k@yahoo.com; 4National Institute of Pharmaceutical Education and Research-Kolkata, Maniktala, Kolkata-700054, India

**Keywords:** leukemic cells, triazole based aminodiol derivative, mitochondrial damage, autophagy, STAT3

## Abstract

Various derivatives that mimic ceramide structures by introducing a triazole to connect the aminodiol moiety and long alkyl chain have been synthesized and screened for their anti-leukemia activity. SPS8 stood out among the derivatives, showing cytotoxic selectivity between leukemic cell lines and human peripheral blood mononuclear cells (about ten times). DAPI nuclear staining and H&E staining revealed DNA fragmentation under the action of SPS8. SPS8 induced an increase in intracellular Ca^2+^ levels and mitochondrial stress in HL-60 cells identified by the loss of mitochondrial membrane potential, transmission electron microscopy (TEM) examination, and altered expressions of Bcl-2 family proteins. SPS8 also induced autophagy through the detection of Atg5, beclin-1, and LC3 II protein expression, as well as TEM examination. Chloroquine, an autophagy inhibitor, promoted SPS8-induced apoptosis, suggesting the cytoprotective role of autophagy in hindering SPS8 from apoptosis. Furthermore, SPS8 was shown to alter the expressions of a variety of genes using a microarray analysis and volcano plot filtering. A further cellular signaling pathways analysis suggested that SPS8 induced several cellular processes in HL-60, including the sterol biosynthesis process and cholesterol biosynthesis process, and inhibited some cellular pathways, in which STAT3 was the most critical nuclear factor. Further identification revealed that SPS8 inhibited the phosphorylation of STAT3, representing the loss of cytoprotective activity. In conclusion, the data suggest that SPS8 induces both apoptosis and autophagy in leukemic cells, in which autophagy plays a cytoprotective role in impeding apoptosis. Moreover, the inhibition of STAT3 phosphorylation may support SPS8-induced anti-leukemic activity.

## 1. Introduction

Leukemia involves various malignant conditions that influence blood and its forming tissues. Normal hematopoiesis is inhibited in leukemia by the uncontrolled production and accumulation of leukemic cells, which are classified according to the types of cells affected and the developmental stages of the initiating cells, and are categorized into acute myeloid leukemia (AML), acute lymphoblastic leukemia (ALL), chronic myeloid leukemia (CML), and chronic lymphoblastic leukemia (CLL) [1,2]. Leukemia is usually considered an uncommon condition. Despite being a low-incidence tumor, leukemia does not equate to low interest. Leukemia remains a key tumor for the rational design of chemotherapeutic approaches and drug discovery [3,4]. However, although significant progress has been made in some types of leukemia, such as CML and CLL, cytotoxic therapy for AML and acute promyelocytic leukemia (APL) has remained more-or-less unaltered in recent decades. AML displays the ability of self-renewal, maintaining malignant populations and likely generating subclones. AML is composed of cancer stem cells that are responsible for disease progression, drug resistance, and relapse after treatment. The detection of intracellular signaling has demonstrated the dysfunction of multiple pathways in AML, which has made researchers rethink current approaches and attracted attention to identifying novel therapeutic targets for AML treatment [5]. 

APL, a unique subtype of AML, is categorized by an abnormal proliferation of immature granulocytes (called promyelocytes), life-threatening coagulopathy, and the hallmark t(15;17)(q24;q21), which generates a PML-RARα (retinoic acid receptor α) fusion gene. The discovery and understanding of APL’s molecular pathogenesis has resulted in clinical therapy using all-trans retinoic acid (ATRA). APL is sensitive to ATRA. ATRA with chemotherapy is the standard therapy for APL, with cure rates exceeding 80% [6]. ATRA does not kill malignant cells directly, but returns APL cells to terminal differentiation and stimulates the degradation of abnormal PML/RARα fusion proteins through proteolytic mechanisms [7,8]. Although it is well tolerated, ATRA therapy may develop severe and life-threatening complications referred to as retinoic acid syndrome, including respiratory distress, dyspnea, fever, pulmonary infiltrates, bone pain, headache, and weight gain [9,10,11]. Retinoic acid syndrome is related to increasing leukocytes counts and is possibly induced by the release of cytokines through maturing blast cells [12]. Accordingly, the prompt administration of steroids (e.g., dexamethasone) at the first sign of unexplained complications is critical [10,11]. Arsenic trioxide, another anti-APL agent, has revolutionized APL treatments in relapsed/refractory disease. Remission with arsenic trioxide has also been reported [13]. Arsenic trioxide induces apoptosis regardless of PML-RARα status, but predominantly through the induction of mitotic arrest of the cell cycle and the mitochondria-mediated intrinsic apoptotic pathway without affecting the architecture of the microtubules [14]. Recently, the success of proteasome inhibitors in multiple myeloma therapy has led to their use in other malignancies, including APL. It has been reported that bortezomib, a proteasome inhibitor, impairs the ubiquitin–proteasome system that regulates protein homeostasis by inducing excessive PML-RARα accumulation, thereby increasing endoplasmic reticulum (ER) stress and killing APL cells [15]. These studies suggest the therapeutic benefit of multiple targeting in the anti-APL mechanism.

The generation of ceramides, long-chain aminodiol sphingosines serving as a lipid backbone, can be induced by chemotherapy, radiation, and oxidative stress, to orchestrate multiple cell responses and cell fates, such as cell cycle arrest, senescence, and cell death [16,17,18]. Accordingly, various derivatives that mimic a ceramide structure by introducing a triazole to connect the aminodiol moiety and long alkyl chain have been synthesized [19]. The screening test for examining cell survival using an MTT (methylthiazolyldiphenyl-tetrazolium bromide) colorimetric assay discovered the active compounds for reducing cell survival in APL HL-60 cells. The anti-APL mechanism and sphingosine kinase-1 activity of the active derivative SPS8 (Figure 1A) were delineated, and the effects on organelle-mediated stress (e.g., ER, autophagosome, and mitochondria) and the receptor-mediated apoptotic pathway (e.g., death receptor) were highlighted. Moreover, transcriptomic profiling was studied to determine the genetic regulation of SPS8 in HL-60 cells, to further realize an anti-APL strategy.

## 2. Results

### 2.1. SPS8 Selectively Induces Cytotoxicity in APL HL-60 Cells

An MTT assay, which relies on a mitochondrial reductase to convert tetrazolium compound to formazan, was applied in this study, to assess the SPS8-induced cytotoxicity in HL-60 and human peripheral blood mononuclear cells (PBMCs). SPS8 induced a time- and concentration-dependent decrease of cell viability in both cell types, with IC_50_ values of 7.19, 5.69, and 1.62 μM in HL-60 and 23.33, 20.10, and 17.19 μM in PBMCs after SPS8 exposure for 24, 48, and 72 hours, respectively (Figure 1B). SPS8 displayed higher activity in HL-60 than that in PBMCs (ranging from 3.24 to 10.61 times). SPS8 also induced a cytotoxic effect in THP-1 (acute monocytic leukemia, one of the types of AML) and MV-4-11 (biphenotypic myelomonocytic leukemia, one of the types of AML) (Appendix A). Dasatinib, a new dual Src/Bcr-Abl tyrosine kinase inhibitor, was used as a reference drug. Dasatinib was initially developed for the treatment of chronic myeloid leukemia (CML). Recently, it has been applied to the treatment of certain APL and AML patients [20,21,22]. The supplementary data revealed the anti-HL-60 activity of dasatinib, with IC_50_ values of 82.78, 59.53, and 4.56 μM after treatment for 24, 48, and 72 hours, respectively (Appendix A). SPS8 showed higher activity than dasatinib, ranging from 2.81 to 11.51 times. DAPI nuclear staining and Giemsa staining demonstrated DNA fragmentation and apoptosis to SPS8 action (Figure 1C). A flow cytometric analysis of DNA staining with propidium iodine (PI) also revealed that SPS8 induced a time-dependent increase of the apoptotic sub-G1 population in HL-60 (Figure 1D), THP-1, and MV-4-11 cells (Appendix A). Furthermore, annexin-V/PI double staining was used to examine the necroptosis effect. The data showed that SPS8 did not induce necroptosis (Appendix A).

### 2.2. SPS8 Induces Mitochondrial Damage in Connecting Apoptotic Signaling Pathways 

The mitochondria, the powerhouse of the cell in energy production, is recognized as a key player in regulating multiple cellular processes, including cell survival and growth, differentiation, metabolism, calcium signaling, and cell death [23]. The mitochondrial function in HL-60 cells was examined by monitoring changes in the mitochondrial membrane potential (ΔΨ_m_), showing that SPS8 induced a time-dependent ΔΨ_m_ loss (Figure 2A). Similar effects were also obtained for both THP-1 and MV-4-11 cells (Appendix A). The transmission electron microscopy (TEM) analysis also demonstrated SPS8-induced mitochondrial damage through the detection of mitochondrial swelling (please see below). Bcl-2 family proteins, which consist of anti-apoptotic (e.g., Bcl-2 and Mcl-1) and pro-apoptotic members (e.g., Bid, Bim, and PUMA), play a critical role in maintaining the integrity of mitochondrial membranes. SPS8 significantly modified the expressions of Bcl-2 family proteins (e.g., the downregulation of Mcl-1, the upregulation of PUMAα and PUMAβ, and the cleavage of Bid into proapoptotic truncated Bid) (Figure 2B), leading to the opening of permeability transition pores and the loss of ΔΨ_m_. Moreover, SPS8 resulted in a dramatic upregulation of cytochrome *c* (Figure 2B), a mitochondrial respiratory chain protein with dual functions in regulating cellular energetic metabolism and apoptosis. It also induced the activation of both intrinsic (mitochondria-involved) and extrinsic (death receptor-involved) apoptotic caspase cascades, including the generation of cleaved caspase-9 and -8 (two initiator caspases) and cleaved caspase-3 (an executioner caspase). The increase in PARP-1 cleavage (a caspase-3 substrate) also was apparently due to SPS8 action (Figure 2C). Moreover, SPS8 induced a profound formation of gamma-H2A.X, an early chromatin modification after the initiation of DNA fragmentation during apoptosis [24]; in contrast, surviving, which is a member of the inhibitor of apoptosis (IAP) protein family in blocking caspases, was downregulated by SPS8 (Figure 2C). Collectively, these results indicate that SPS8-induced apoptosis is involved in mitochondrial damage and, most likely, interactions with the extrinsic apoptosis pathway, leading to the activation of caspase cascades and ultimately cell death.

### 2.3. SPS8 Induces Autophagy in HL-60 Cells

Autophagy is a cellular recycling process that degrades and removes unnecessary or stressed cytoplasmic organelles, proteins, and macromolecules (as well as the recycling of breakdown products) [25]. Autophagy is a crucial cellular program with dual roles in promoting either cell survival or cell death [26,27]. LC3, which acts in substrate selection and autophagosome biogenesis in autophagy, is a widely recognized autophagosome marker. P62 is a protein adaptor in the autophagic proteolytic processes in regulating the sequestration of toxic, aggregate-prone cargo proteins [28,29]. The autophagy in this study was examined using the immunofluorescence detection of LC3. The data demonstrated autophagy induction by SPS8 exposure (Figure 3A). The TEM examination revealed the SPS8-induced formation of autophagosome-like structures containing double membranes and engulfed organelle debris (Figure 3B). Some of the structures were swelling mitochondria (Figure 3B) after SPS8 exposure, that were suggestive of an induced rupture of the mitochondrial outer membrane and the release of proapoptotic proteins, such as cytochrome *c*. Several reliable autophagy markers were examined using Western blotting analysis. The data in Figure 3C demonstrate that SPS8 induced a time-dependent increase in the protein expression of Atg5 and LC3 II. In contrast, SPS8 induced a time-dependent change of p62 protein expression in a biphasic manner (Figure 3C). Collectively, the data substantiate the induction of autophagy to SPS8 action. 

Autophagy that maintains cellular homeostasis through reutilizing certain intracellular organelles and breakdown products may play a pro-survival role during cellular stress; nevertheless, autophagy can induce cell death under some conditions [30,31]. Chloroquine, a classic autophagy inhibitor [32], suppresses the fusion of autophagosomes with lysosomes and lysosomal protein degradation, by altering the acidic environment of lysosomes. A cytoflowmetric analysis with PI DNA staining showed that chloroquine did not significantly promote a SPS8-induced increase of the apoptotic sub-G1 population until 24 hours of treatment was achieved (Appendix A). The presence of chloroquine also potentiated the SPS8-induced activation of caspase-9 and -3 and encouraged γ-H2A.X formation (Appendix A). The data suggest that autophagy may play a cytoprotective role in hindering SPS8 from cell apoptosis after a longer treatment time (e.g., 24 hours).

### 2.4. Calcium Plays a Role in the SPS8-Mediated Effect in HL-60 Cells

The factors that govern the interplay between apoptosis and autophagy are not clearly understood. Recent advances have investigated several crucial players and their complex interaction pathways, including mTOR, Beclin-1, Bcl-2 family proteins, Atg family proteins, and IAPs [30,31]. Notably, intracellular calcium, even with a slight alteration in frequency or amplitude through various stimuli, can result in apoptosis and autophagy [33]. In the present work, a flow cytometric analysis was applied in HL-60 cells to monitor intracellular Ca^2+^ levels with Fluo-3 staining. The data showed that SPS8 induced an increase of intracellular Ca^2+^ levels in a concentration-dependent manner (Figure 4). The intracellular calcium chelator BAPTA was shown to significantly protect the cells from SPS8-induced cytotoxic effects using an MTT assay, suggesting a functional role of the calcium signaling pathway during SPS8 action.

### 2.5. SPS8 Exhibits Transcriptional Regulation in HL-60 Cells

The microarray platform Affymetrix HG-U133A plus 2 was applied in this study to determine the regulated transcription of the genes in HL-60 cells under SPS8 treatment. Volcano plot filtering was used to quickly identify the genes with altered expression levels between the SPS8-treated and vehicle-treated groups in large data sets (Figure 5A). The data demonstrated that SPS8 induced a dramatic upregulation of gene expression in 50 genes and downregulation in nine genes (fold change ≥ 2.0, *p* < 0.05) (Table 1). A GeneGo Pathway Map analysis from the MetaCore integrated software was used to analyze the enrichment genes in functional ontologies to propose the possible metabolic pathways that the regulated genes might be involved in. The data for the SPS8-induced gene upregulation suggest several cellular processes, including the sterol biosynthesis process, cholesterol biosynthesis process, sterol metabolic process, steroid biosynthesis process, and cholesterol metabolic process. In contrast, the data on SPS8-induced gene downregulation suggest the inhibition of several cellular pathways or processes, including cellular responses to drugs, neutrophil homeostasis, neutrophil chemotaxis, myeloid cell homeostasis, leukocyte migration, and leukocyte chemotaxis (Figure 5B). The gene network was built based on the most strongly affected process, “cellular responses to drugs”, and several crucial factors were proposed (Figure 6A). Among them, the Signal Transducer and Activator of Transcription 3 (STAT3) was the most crucial nuclear factor. STAT3, a member of the STAT family proteins, is phosphorylated by receptor-associated Janus kinases (JAK) in response to various stimuli, leading to nuclear translocation and transcriptional activation for inflammation, cell survival, proliferation, metastasis, and immune evasion [34]. Notably, it has been reported that STAT3 fragmentation can be mediated by caspases, leading to a decrease in transcriptional activity [35]. The data in Figure 6B reveal that SPS8 inhibits the phosphorylation of STAT3 and induces a profound cleavage of STAT3 that might be attributable to caspase activation. The data also indicate the decline of STAT3 activity to SPS8 action.

## 3. Discussion

APL patients were threatened by a high risk of early mortality in the past because of severe coagulopathy, frequently resulting in lethal cerebral hemorrhaging. APL has become one of the most remediable subtypes of AML after the therapeutic initiation of the differentiating agent, ATRA. However, disease relapse remains a challenging obstacle in ATRA treatment. Accordingly, several treatment options are available for APL patients who have relapsed after ATRA, including arsenic trioxide and cytarabine [36]. Molecular target drugs for introducing caspase-dependent apoptosis in leukemia cells have demonstrated promising activity in existing drugs (e.g., arsenic trioxide and cytarabine) and in the drugs in clinical trials [37], and have encouraged the discovery of new molecular target drugs for APL/AML therapy. However, dose-limiting toxicity is one of the major challenges for these therapeutic drugs. In the present work, SPS8 displayed higher activity in HL-60 than that in PBMCs, particularly under long-term treatment, with a more than ten-fold selectivity, suggesting the potential for further development. 

Mitochondria-involved intrinsic apoptosis and death receptor-mediated extrinsic apoptosis frequently engage in crosstalk in the caspase-dependent apoptotic cell death program [38,39]. Our data suggest the involvement of both intrinsic and extrinsic apoptosis pathways in SPS8 action. Notably, SPS8 induced the early and time-dependent cleavage of Bid, a caspase-8 substrate, into truncated Bid (tBid). It has been suggested that tBid can translocate to the mitochondria, inserting into the mitochondrial outer membrane and resulting in the release of cytochrome *c* to promote the formation of a cytochrome *c*/Apaf-1/pro-caspase-9 complex, that induces the activation of caspase-9 and downstream executioner caspase-3 and -7 [40]. Alternatively, tBid has been suggested to function as a membrane-targeted death ligand for binding to and allosterically activating Bak, leading to cytochrome *c* release [41]. Collectively, the data indicate that death receptor-mediated apoptosis signaling might play a key role in connecting the mitochondria-involved apoptosis pathway to SPS8 action. In interpreting the death receptor-mediated pathway, the apoptotic protein array was applied. The data demonstrate that SPS8 induced an increase of protein expressions for various death receptors and ligands, including TNF-α, FasL (Fas ligand), TRAILR1 (also known as death receptor 4), TRAILR2 (also known as death receptor 5), and DR6 (death receptor 6, also known as the tumor necrosis factor receptor superfamily member 21, TNFRSF21) (Appendix A), suggesting the contribution of the death receptor-mediated pathway. In recent decades, Bim and PUMA have been documented to play a key role in apoptotic initiation during apoptotic stimuli in a wide variety of cell types, in particular those of haemopoietic origin [42,43]. Furthermore, the anti-apoptotic Bcl-2 family member, Mcl-1, plays a critical role in the development and maintenance of AML, making it a potential target in this disease [44]. Our data show that SPS8 induced the upregulation of PUMA protein expression but a decrease in Mcl-1, suggesting the contribution of these two Bcl-2 family members to the mitochondrial pathway of apoptosis.

Apoptosis and autophagy are two crucial cellular processes that determine a cell’s fate. It has been demonstrated well that apoptosis does not act by itself, but in a complicated interplay with autophagy to determine a cell’s fate [31,45]. Autophagy can collaborate with apoptosis to induce cell death or can function as a cell survival promotor in suppressing apoptosis. Moreover, the molecular players of both cellular processes are interconnected [45]. The autophagy inhibitor, chloroquine, was used to examine the relationship between apoptosis and autophagy. As a result, chloroquine significantly promoted SPS8-induced apoptosis until 24 hours of treatment was achieved, suggesting the cytoprotective role of autophagy in hindering SPS8 from cell apoptosis after a longer treatment time. An autophagy-mediated cytoprotective effect was also reported in leukemic cells in response to a variety of apoptotic stimuli. Accordingly, it has been suggested that the inhibition of autophagy might provide new therapeutic options for AML treatment [46].

Cytoplasmic Ca^2+^ has been highlighted as a key regulator for fine-tuning apoptotic and autophagic responses. Various Ca^2+^-dependent kinases and signaling molecules have been identified as important executioners in both cell apoptosis and autophagy [33]. Our data reveal that the SPS8-induced increase of intracellular Ca^2+^ levels play a functional role, since the chelation of intracellular calcium by BAPTA significantly rescued cell viability. Our data also show that the protein levels of both Atg5 and Beclin-1 were, albeit insignificantly, decreased during SPS8 exposure for 15 and 24 hours. This is possibly due to intracellular Ca^2+^ elevation and caspase activation, since Atg5 and Beclin-1 have been identified to be substrates of calpain and caspase-3, respectively [47,48]. These studies also suggest that the cleavage of Atg5 and Beclin-1 can switch autophagy to apoptotic cell death [47,48]. However, our data show that autophagy is still apparent, based on observing high LC3 II levels and autophagosome-like structures under TEM examination. Similar to our data, it has been reported that Beclin-1 silence not only increases the autophagic process, but also decreases apoptosis in gemcitabine-treated pancreatic cancer MIA PaCa-2 cells [49]. Therefore, the regulation between complex cellular molecules during autophagy needs further elucidation.

The data from the analysis of the gene expression arrays show that SPS8 induced a marked increase of the expressions of several genes, such as HMGCS1, INSIG1, MSMO1, DHCR7, HMGCR, FADS1, STARD4, SQLE, LSS, IDI1, and FDFT1, which are related to the processes of sterol biosynthesis, cholesterol biosynthesis, sterol metabolism, steroid biosynthesis, cholesterol transport, and cholesterol metabolism. Cholesterol, a key component of the cell membrane, functions as a precursor for a variety of bioactive molecules. Cholesterol biosynthesis, which requires the coordinated activity of dozens of enzymes, is transcriptionally orchestrated by the master regulator, sterol-regulatory element–binding proteins (SREBPs) [50,51]. The decrease of cholesterol content in the endoplasmic reticulum (ER) membrane induces multiple processes to release and cleave SREBPs into an active N-terminal domain, that translocates to the nucleus and triggers the expression of multiple related genes [52]. Cholesterol biosynthesis is vulnerable to various feedback pathways, which guarantees that cholesterol production satisfies cellular needs. Many of the SPS8-induced genes are SREBP target genes, suggesting the activation of SREBP. Moreover, a crucial role of ER stress and Ca^2+^ has been suggested in determining the susceptibility of the sterol sensing mechanism essential to the SREBP activation pathway [53,54]. Our data show the capability of SPS8 in stimulating intracellular Ca^2+^ mobilization and ER stress, suggesting its potential in SREBP activation and the subsequent modification of sterol biosynthesis and metabolism.

The alternative splicing of STAT3 generates two isoforms: STAT3α and STAT3β. The activation of STAT3α plays a crucial role in stimulating the oncogenic signaling pathways, including cell proliferation and survival, whereas STAT3β is considered to be a dominant–negative regulator of cancer [55]. Recently, STAT3β has been suggested to be a significant transcriptional regulator that cross-controls STAT3α’s functions [56]. Multiple lines of evidence suggest that STAT3 plays a central role in neutrophil chemotaxis and mobilization and large B-cell lymphoma dissemination [57,58]. Constitutive STAT3 activation has been apparent in a variety of hematopoietic and non-hematopoietic malignancies, where it is responsible for cell metastasis, survival, and apoptotic resistance. Recently, it has been reported that the inhibition of STAT3 can amplify arsenic trioxide-mediated anti-AML activity [59]. Furthermore, APL fusion proteins in STAT3 signaling are suggested to contribute to apoptotic resistance, indicating that STAT3 may be a potential target for anti-APL strategies [60]. The present study show that SPS8 facilitates STAT3 inhibition through its cleavage. These data support the SPS8-mediated inhibition of cell migration and chemotaxis activity proposed by the analysis of the gene expression microarray data. Moreover, STAT3 inhibition could also support SPS8-induced cytotoxic effects and the apoptotic signaling pathway in APL cells. 

In conclusion, the data suggest that SPS8 is an effective anti-APL agent, with a lesser effect on human PBMCs. SPS8 induces a cytotoxic effect and apoptotic signaling cascades in a sequential manner (Figure 7). It induces the activation of both mitochondria-involved intrinsic and death receptor-mediated extrinsic apoptosis pathways, in which tBid serves as a crosstalk regulator that assists in the mitochondrial dysfunction caused by Mcl-1 downregulation and PUMA upregulation. Increased intracellular Ca^2+^ mobilization may contribute to both apoptosis and autophagy. Moreover, the suppression of STAT3 phosphorylation and the induction of STAT3 cleavage also supports SPS8-induced anti-APL activity.

## 4. Materials and Methods

### 4.1. Chemicals, Reagents, Antibodies and Cell Culture

MTT, dimethyl sulfoxide (DMSO), PI, dasatinib, and necrostatin-1 were purchased from Sigma-Aldrich (St. Louis, MO). JC-1 (5,5’,6,6’-tetrachloro-1,1’,3,3’-Tetraethylbenzimidazolyl carbocyanine iodide) was obtained from Molecular Probes (Invitrogen, Germany). The antibodies of Bcl-2, GAPDH, PUMA, Bim, Mcl-1, α-tubulin, PARP-1, and p62 were obtained from Santa Cruz Biotechnology, Inc. (Santa Cruz, CA, USA). Antibodies of Bid, cleaved caspase-9, caspase-8, γH2A.X^Ser139^, and survivin were obtained from Cell Signaling Technologies (Boston, MA). LC3 was obtained from Genetex (Irvine, CA, USA). Cytochrome *c*, beclin, and STAT3 were obtained from *BD* Biosciences (San Jose, CA, USA). Anti-mouse and anti-rabbit IgGs were obtained from Jackson Immuno-Research Laboratories, Inc. (West Grove, PA, USA). HL-60 and THP-1 were obtained from the Bioresource Collection and Research Center (BCRC, Taiwan). MV-4-11 was obtained from American Type Culture Collection (Rockville, MD, USA). Human PBMCs were purchased from Zen-Bio Inc. (Research Triangle Park, NC, USA). All cells were grown in a water-saturated atmosphere under 5% CO_2_ at 37 °C. HL-60 cells were cultured in Iscove’s Modified Dulbecco’s Medium (IMDM) (Gibco, Grand Island, NY, USA) with 10% FBS (*v*/*v*) and penicillin (100 U/mL)/streptomycin (100 μg/mL). PBMCs were cultured in an RPMI-1640 Medium (Gibco, Grand Island, NY, USA) with 10% FBS (*v*/*v*) and penicillin (100 U/mL)/streptomycin (100 μg/mL).

### 4.2. Cell Viability Assay

The cells, at a density of 3 × 10^5^ cells/mL, were cultured at a 24-well plate. After the aforementioned treatment, the mitochondrial MTT reduction activity was assessed. MTT was dissolved in phosphate-buffered saline (PBS), at a concentration of 5 mg/mL and filtered. The stock MTT solution was added to the medium of each well (final concentration, 0.5 mg/mL) and the plates were incubated at 37 °C for 2 h. Then, the cells were lysed with 100% DMSO, and the plate was read by an enzyme-linked immunosorbent assay (ELISA) reader (570 nm), to obtain the absorbance density values.

### 4.3. Flow Cytometric Detection of Apoptosis

After the treatment, the cells were harvested, washed with ice-cold PBS, fixed with 70% ethanol at 4 °C for 30 min, and washed with ice-cold PBS. The cells were resuspended with a PI solution (0.3 mL) containing Triton X-100 (0.1% *v*/*v*), RNase (100 mg/mL) and PI (80 mg/mL) in the dark, and were analyzed with the FACScan and CellQuest software (Becton Dickinson, Mountain View, CA, USA). The population of apoptosis that contains sub-G1 DNA content was obtained as a percentage of the total number of cells. Furthermore, the apoptosis was determined using the staining with FITC-Annexin V/PI by an apoptosis detection kit (BD Pharmingen). The experiment was performed according to the manufacturer’s protocol. The apoptosis was detected and quantified using a flow cytometric analysis (*BD* Biosciences).

### 4.4. Measurement of ΔΨm

Cells were treated with or without the indicated agent. Thirty minutes before the termination of incubation, the cells were incubated with JC-1 (final concentration of 2 μM) at 37 °C for 30 min. The cells were harvested and the accumulation of JC-1 was examined using a flow cytometric analysis (*BD* Biosciences).

### 4.5. Confocal Microscopic Examination with DAPI Staining

After the indicated treatment, the cells were fixed (100% methanol) at −20°C for 5 min and incubated in DAPI solution (1 μg/mL) for nuclear staining or in the indicated antibodies for the detection of target proteins. The cells were analyzed by a confocal laser microscopic system (Leica TCS SP2).

### 4.6. Microscopic Observation of Cell Morphology

After the indicated treatment, the cells were collected by centrifugation and resuspended in PreserveCyt solution (200 mL, PBS plus methanol, Hologic Inc., Marlborough, MA, USA). The cell suspension, passed through a Thinprep processing machine (Hologic Inc., Marlborough, MA, USA), was collected. The slides fixed in alcohol (95%) and stained with Wright-Giemsa (5 min, room temperature) were examined under a fluorescence microscope (Olympus Corp., MA, USA).

### 4.7. Transmission Electron Microscopy

After the indicated treatment, the cells were harvested and washed with PBS, and the cell pellets were consecutively fixed in glutaraldehyde (2.5%) and osmium tetroxide (1%). The cells were dehydrated using alcohol and embedded in epoxy resin for sectioning. Ultra-thin sections stained with uranyl acetate and lead citrate were examined with a Hitachi H-7500 transmission electron microscope (Fukuoka, Japan).

### 4.8. Measurement of Intracellular Ca^2+^ Level

After incubation with fluo-3/AM (5 μM) for 30 min, the cells were washed twice and incubated in a fresh medium. A vehicle (0.1% DMSO) or SPS8 was added to the cells, and the intracellular Ca^2+^ level was measured by a flow cytometric analysis (*BD* Biosciences).

### 4.9. RNA Extraction, Microarray and Data Processing

Total RNA were isolated from the cells using a Trizol reagent (Invitrogen, Carlsbad, CA, USA), as recommended by the manufacturer, followed by RNA cleanup using a MinElute Kit (Thermo Fisher Scientific Inc., Waltham, MA, USA). RNA quality was assessed using an Agilent 2100 Bioanalyzer (Agilent Technologies, Inc., Santa Clara, CA, USA). The first-strand cDNA was synthesized from 2 μg of total RNA, using the Super-Script first-strand synthesis system (Invitrogen, Carlsbad, CA, USA). Then, 15 μg of fragmented cRNA was hybridized to each Affymetrix HG-U133A plus 2 Array (Affymetrix, Santa Clara, CA, USA). RNA labeling, hybridization, washing, and processing were performed (Genomic Medicine Research Core Laboratory, Chang Gung Memorial Hospital, Taiwan). Microarray gene profiles were obtained from the vehicle (0.1% DMSO) or SPS8-treated sample. Expression genes with significant differences were analyzed (group comparison). Those presenting statistical significance (with a *p* value < 0.05) and expressions with two-fold or greater difference between the two groups were selected for further analysis. A biological function analysis of the gene-encoded proteins was performed using a MetaCore analysis (Life Sciences Research, Thomson Reuters, UK). The pathway analysis and additional analyses of gene functions were determined using gene ontology (GO) analysis and/or the Kyoto Encyclopedia of Genes and Genomes (KEGG) pathway analysis (Kyoto, Japan).

### 4.10. Human Apoptosis Antibody Array

Apoptosis-related proteins were determined using an antibody array (human apoptosis antibody array kit, Raybiotech, Norcross, GA, USA). After the indicated treatment, the cells were collected in which the protein extract (300 μg) from each sample was loaded onto an antibody array membrane for 4 h. Each membrane was quantified using a Biospectrum AC ChemiHR 40 system, and the membrane image file was analyzed using a UVP analysis software (UVP, Upland, CA, USA).

### 4.11. Western Blotting

After the indicated treatment, the cells were harvested and lysed in a lysis buffer (0.1 mL), containing 20 mM Tris–HCl (pH 7.4), 150 mM NaCl, 1% Triton X-100, 1 mM EDTA, 1 mM PMSF, 1 μg/ml leupeptin, 1 mM NaF, 1 mM sodium orthovanadate, and 1 mM dithiothreitol. After the quantification and mixing with the sample buffer, the protein was boiled (95 °C, 5 min). The amount of protein (30 μg) was separated by electrophoresis in SDS-PAGE, transferred to PVDF membranes and detected through conjugation with specific antibodies. After being labeled with secondary antibody, the immunoreactive proteins were detected using an enhanced chemiluminescence detection kit (Amersham, Buckinghamshire, UK).

### 4.12. Statistical Analysis 

Data are presented as the mean ± SEM with the indicated number of independent experiments. A statistical analysis was performed and two-group comparisons were done with Student’s *t*-test. *p* < 0.05 was considered statistically significant.

## Figures and Tables

**Figure 1 ijms-21-02470-f001:**
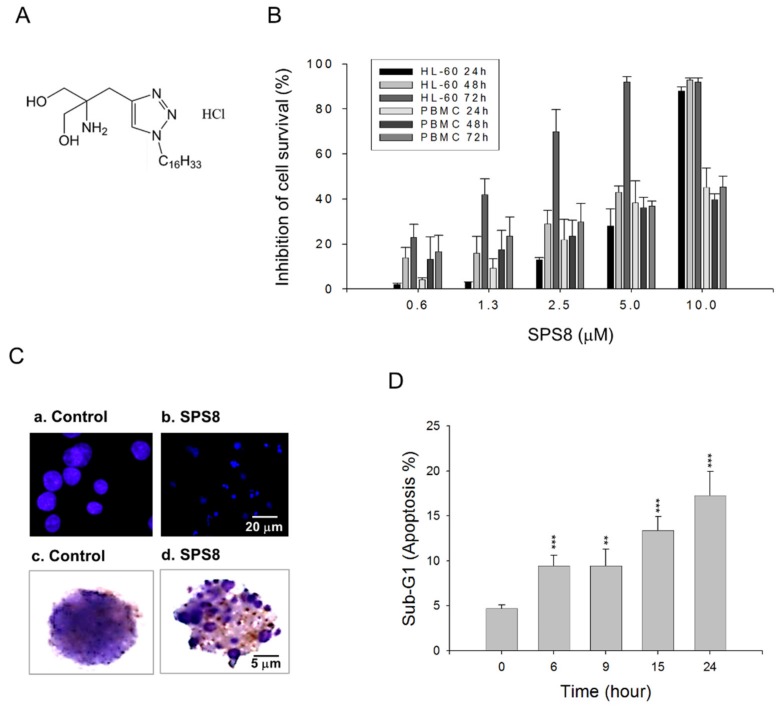
Effect of SPS8 on cell viability of HL-60 cells. (**A**) Chemical structure of SPS8. (**B**) Graded concentrations of SPS8 were added to HL-60 or human peripheral blood mononuclear cells (PBMCs, purchased from Zen-Bio Inc. Research Triangle Park, NC, USA) for 24, 48, or 72 h. The cytotoxic effects were determined by an MTT assay. (**C**) HL-60 cells were incubated in the absence or presence of SPS8 (5 μM) for 24 h. DAPI nuclear staining (a,b) and Giemsa staining (c,d) were performed. (**D**) HL-60 cells were incubated in the absence or presence of SPS8 (5 μM) for the indicated times. A flow cytometric analysis of DNA staining with propidium iodine was performed. Data are expressed as the mean ± SEM of three independent experiments. ** *p* < 0.01 and *** *p* < 0.001 compared with the control.

**Figure 2 ijms-21-02470-f002:**
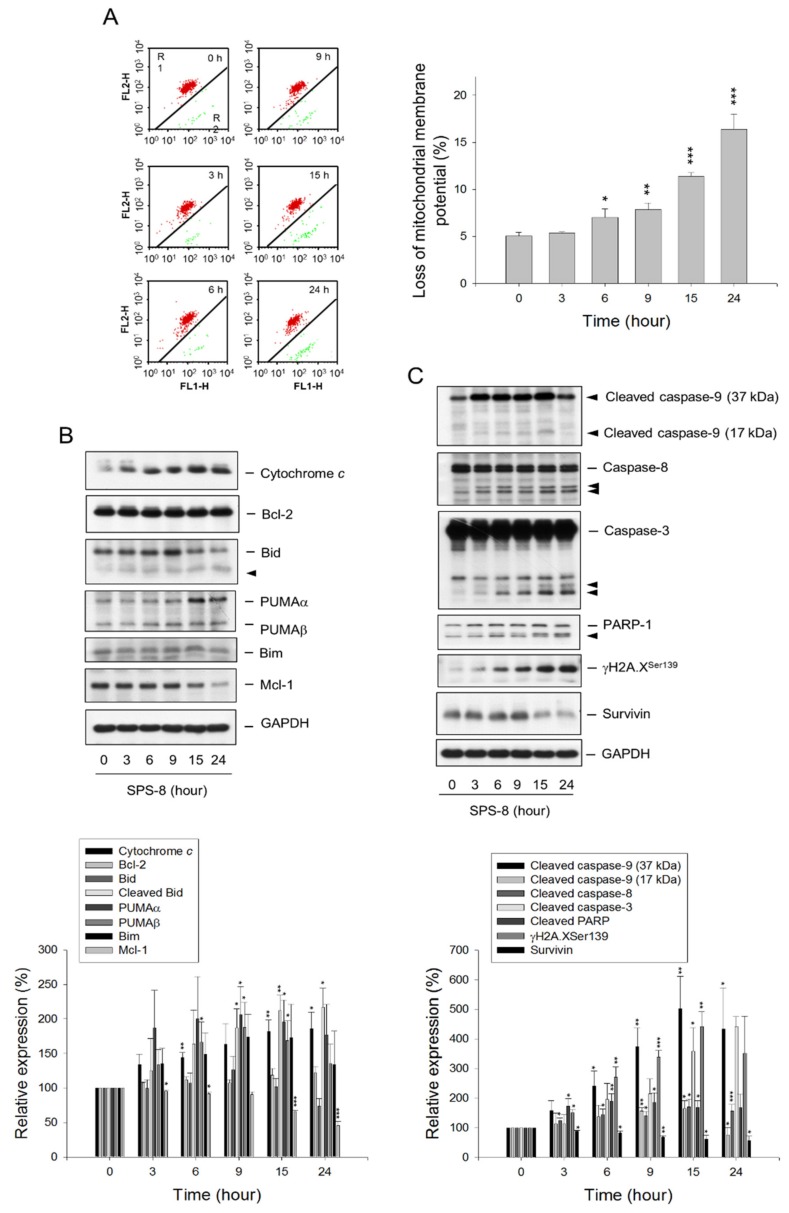
Effect of SPS8 on mitochondrial membrane potential and related protein expression. HL-60 cells were incubated in the absence or presence of SPS8 (5 μM) for the indicated times. (**A**) The cells were incubated with JC-1 dye for the detection of mitochondrial membrane potential using a flow cytometric analysis. Data are expressed as the mean ± SEM of three independent experiments. * *p* < 0.05, ** *p* < 0.01 and *** *p* < 0.001 compared with the zero time control. (**B**,**C**) The cells were harvested for the detection of protein expression using Western blotting. The expressions were quantified using the Image Lab Software 6.0 (BIO-RAD). Data are expressed as the mean ± SEM of three to six independent experiments. * *p* < 0.05, ** *p* < 0.01 and *** *p* < 0.001 compared with the control.

**Figure 3 ijms-21-02470-f003:**
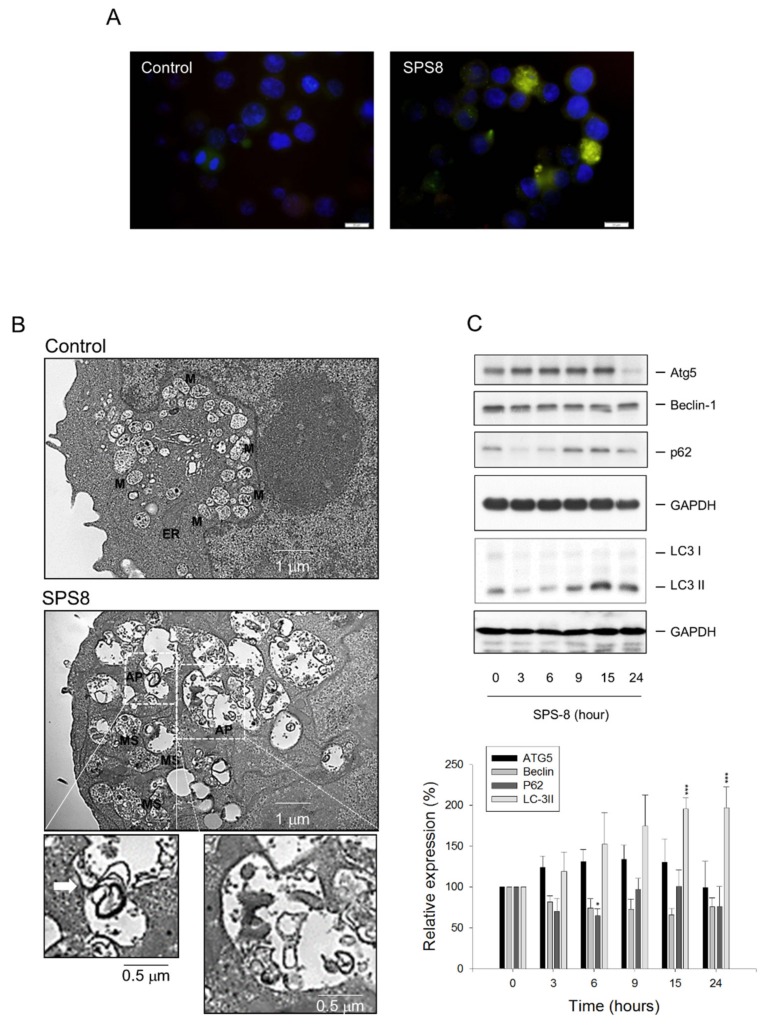
Effect of SPS8 on the induction of autophagy. HL-60 cells were incubated in the absence or presence of SPS8 (5 μM) for 24 h (**A**, **B**) or the indicated times (**C**). (**A**) The immunofluorescence staining for LC3 (green) and DAPI (blue) was performed. Scale bar, 20 μm. (**B**) A TEM analysis of organelle morphology was performed. The endoplasmic reticulum (ER), mitochondria (M), and swelling mitochondria (SM) were observed. The formation of autophagosome (AP)-like structures containing double membranes and engulfed organelle debris was also detected. (**C**) The cells were harvested for the detection of protein expression, using Western blotting. Several autophagy-related protein expressions were examined. The expressions were quantified using Image Lab Software 6.0 (BIO-RAD). Data are expressed as the mean ± SEM of four to six independent experiments. * *p* < 0.05 and *** *p* < 0.001 compared with the zero time control.

**Figure 4 ijms-21-02470-f004:**
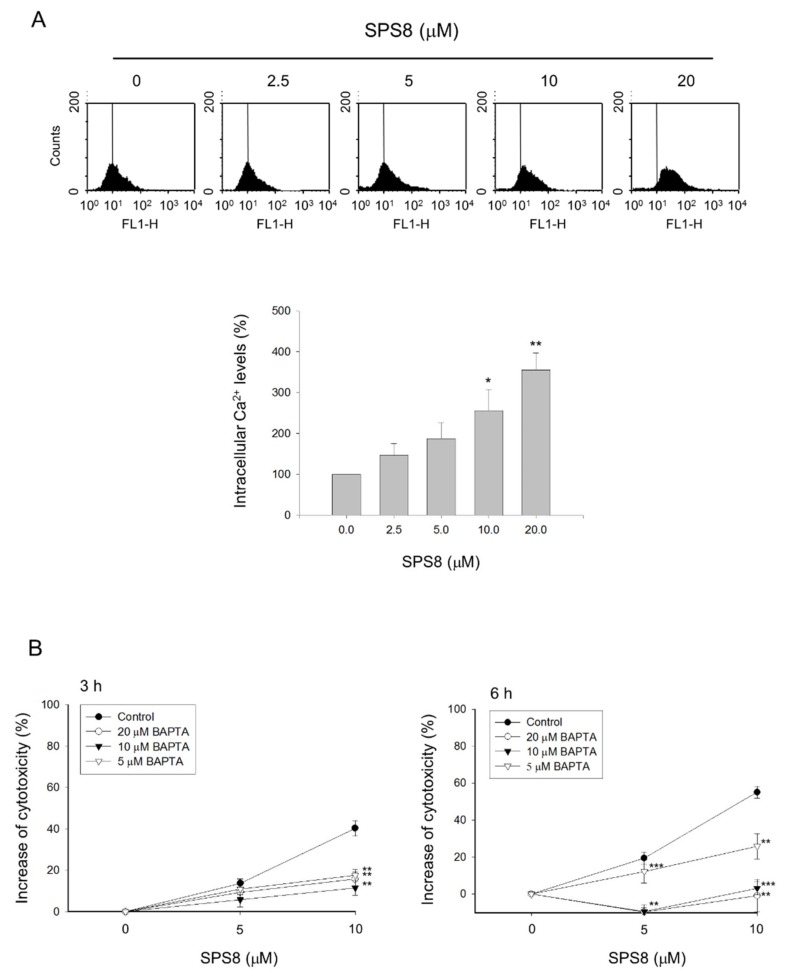
Effect of SPS8 on intracellular calcium levels of HL-60 cells. (**A**) HL-60 cells were incubated in the absence or presence of SPS8 at the indicated concentrations for 30 min. The intracellular Ca^2+^ levels were measured by a flow cytometric analysis using fluo-3/AM staining. *** *p* < 0.001 compared with zero micromolar control. (**B**) HL-60 cells were incubated in the absence or presence of the indicated agent for 3 or 6 h, and the cytotoxic effect was determined by an MTT assay. Data are expressed as the mean ± SEM of three independent experiments. * *p* < 0.05, ** *p* < 0.01 and *** *p* < 0.001 compared with the BAPTA-free control.

**Figure 5 ijms-21-02470-f005:**
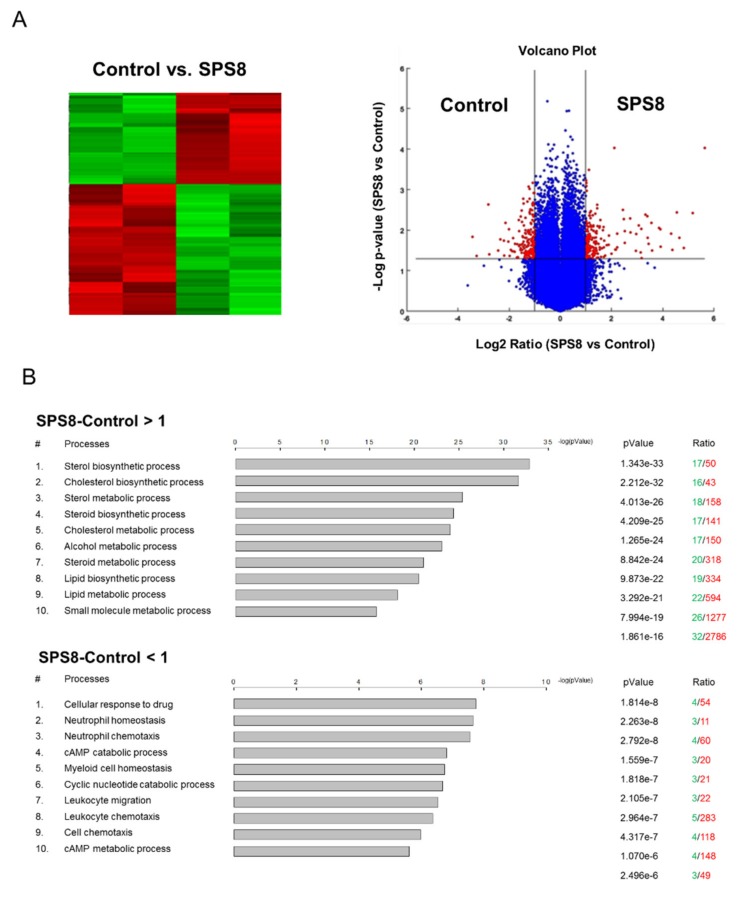
Differentially expressed genes and pathways deregulated in HL-60 cells treated with SPS8. (**A**) A volcano plot and heat map of the expression profiles of genes deregulated in cells treated with SPS8 (5 μM, 6 h), compared with untreated cells. Red denotes high expression; green indicates low expression. (**B**) List of pathways identified by the gene ontology (GO) enrichment analysis. Top 10 networks of up-regulated (SPS8-Control >1) and down-regulated (SPS8-Control < 1) genes that were differentially expressed in the cells treated with SPS8 compared to the control.

**Figure 6 ijms-21-02470-f006:**
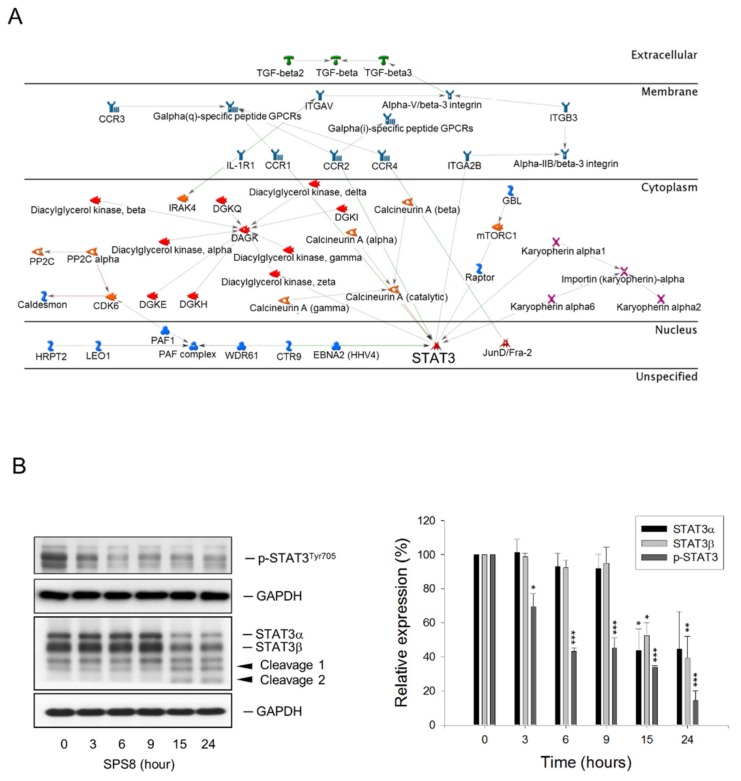
Proposed molecular pathways for SPS8 action. (**A**) The biological interactions of STAT3 co-expressed genes were analyzed by using Metacore. Red line, negative/inhibition effects; Green line, positive/activation effects. Gray line, unspecified effects. (**B**) Western blot analysis of phosphorylated STAT3 and STAT3 α/β levels in HL-60 cells treated with SPS8 (5 μM) for the indicated times. Data are expressed as the mean ± SEM of three independent experiments. * *p* < 0.05, ** *p* < 0.01, and *** *p* < 0.001 compared with the control.

**Figure 7 ijms-21-02470-f007:**
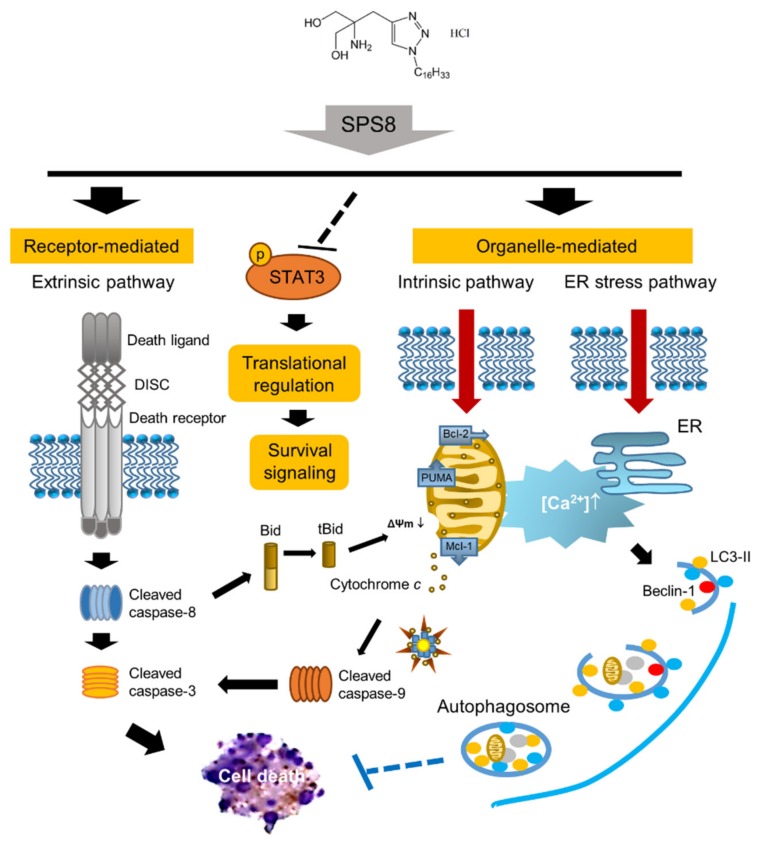
SPS8-mediated apoptotic signaling cascades in HL-60 cells. SPS8 induces the activation of both mitochondria-involved intrinsic and death receptor-mediated extrinsic apoptosis pathways, in which tBid serves as a crosstalk regulator, assisting the mitochondrial dysfunction caused by Mcl-1 downregulation/PUMA upregulation. Increased intracellular Ca^2+^ mobilization contributes to both apoptosis and autophagy. However, autophagy may play a cytoprotective role in hindering SPS8 from cell apoptosis after a longer treatment time. Moreover, SPS8 can also suppress the phosphorylation of STAT3, thereby inhibiting the activation of cellular survival signaling.

**Table 1 ijms-21-02470-t001:** SPS8 induced a dramatic upregulation of gene expression in 50 genes and a downregulation in nine genes (fold change ≥ 2.0, *p* < 0.05) using microarray platform Affymetrix HG-U133A plus 2 analysis.

Up-Regulated Genes		
**Gene Title**	**Gene Symbol**	**log2ratio**
3-hydroxy-3-methylglutaryl-CoA synthase 1 (soluble)	*HMGCS1*	3.870
insulin induced gene 1	*INSIG1*	3.212
methylsterol monooxygenase 1	*MSMO1*	2.935
7-dehydrocholesterol reductase	*DHCR7*	2.759
cyclin G2	*CCNG2*	2.696
3-hydroxy-3-methylglutaryl-CoA reductase	*HMGCR*	2.639
TSC22 domain family, member 3	*TSC22D3*	2.618
low density lipoprotein receptor	*LDLR*	2.503
fatty acid desaturase 1	*FADS1*	2.486
StAR-related lipid transfer (START) domain containing 4	*STARD4*	2.444
squalene epoxidase	*SQLE*	2.437
lanosterol synthase (2,3-oxidosqualene-lanosterol cyclase)	*LSS*	2.386
solute carrier organic anion transporter family, member 4C1	*SLCO4C1*	2.305
tubulin, alpha 1a	*TUBA1A*	2.275
isopentenyl-diphosphate delta isomerase 1	*IDI1*	2.257
Farnesyl-diphosphate farnesyltransferase 1	*FDFT1*	2.200
RUSC1 antisense RNA 1 (non-protein coding)	*RUSC1-AS1*	2.074
sterol-C5-desaturase (ERG3 delta-5-desaturase homolog, S. cerevisiae)-like	*SC5DL*	1.802
acyl-CoA synthetase short-chain family member 2	*ACSS2*	1.797
stearoyl-CoA desaturase (delta-9-desaturase)	*SCD*	1.790
acetyl-CoA acetyltransferase 2	*ACAT2*	1.652
chloride channel, voltage-sensitive 6	*CLCN6*	1.641
24-dehydrocholesterol reductase	*DHCR24*	1.607
methylenetetrahydrofolate reductase (NAD(P)H)	*MTHFR*	1.603
arrestin domain containing 3	*ARRDC3*	1.582
hydroxysteroid (17-beta) dehydrogenase 7	*HSD17B7*	1.565
syntaxin binding protein 1	*STXBP1*	1.522
chromosome 14 open reading frame 1	*C14orf1*	1.512
mevalonate kinase	*MVK*	1.511
Kruppel-like factor 6	*KLF6*	1.481
isocitrate dehydrogenase 1 (NADP+), soluble	*IDH1*	1.411
mevalonate (diphospho) decarboxylase	*MVD*	1.379
sialidase 1 (lysosomal sialidase)	*NEU1*	1.371
IDI2 antisense RNA 1 (non-protein coding)	*IDI2-AS1*	1.325
family with sequence similarity 117, member A	*FAM117A*	1.322
Kruppel-like factor 7 (ubiquitous)	*KLF7*	1.296
lipin 1	*LPIN1*	1.294
methyltransferase like 7A	*METTL7A*	1.224
RAB33A, member RAS oncogene family	*RAB33A*	1.163
centromere protein I	*CENPI*	1.159
jun proto-oncogene	*JUN*	1.148
MCM3AP antisense RNA 1 (non-protein coding)	*MCM3AP-AS1*	1.129
cytoplasmic FMR1 interacting protein 2	*CYFIP2*	1.127
transmembrane 7 superfamily member 2	*TM7SF2*	1.119
Kruppel-like factor 2 (lung)	*KLF2*	1.094
kelch-like 24 (Drosophila)	*KLHL24*	1.092
pantothenate kinase 3	*PANK3*	1.070
NAD(P) dependent steroid dehydrogenase-like	*NSDHL*	1.059
ELOVL fatty acid elongase 6	*ELOVL6*	1.050
transmembrane protein 97	*TMEM97*	1.019
**Down-Regulated Genes**		
**Gene Title**	**Gene Symbol**	**log2ratio**
myosin regulatory light chain interacting protein	*MYLIP*	−1.036
phosphodiesterase 4B, cAMP-specific	*PDE4B*	−1.052
regulator of G-protein signaling 18	*RGS18*	−1.092
T cell receptor delta variable 3	*TRDV3*	−1.116
SLAM family member 8	*SLAMF8*	−1.167
C-type lectin domain family 5, member A	*CLEC5A*	−1.177
ankyrin repeat domain 22	*ANKRD22*	−1.356
Fc fragment of IgA, receptor for	*FCAR*	−1.432
chemokine (C-C motif) ligand 2	*CCL2*	−1.803

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
