# Peer review of "Mechanistic Study of Triazole Based Aminodiol Derivatives in Leukemic Cells—Crosstalk between Mitochondrial Stress-Involved Apoptosis and Autophagy"

_ijms, 2020, doi:10.3390/ijms21072470_

Round 1
Reviewer 1 Report
Acute Promyelocytic Leukemia (APL) are hematological malignancies who need new effective therapeutic options.
The work presented here is focused on the mechanism of action of triazole based aminodiol derivatives against APL and particularly on the promising SPS8 derivative.
The authors report that sp8 induces the biosynthesis of sterol and cholesterol as well as the cleavage of the cytoprotective molecule STAT3. They also report that SPS8 stimulates both autophagic and apoptotic events.
The article is extremely well written and the figures are of very high quality. The authors also took care to show the original blots. The results are very interesting and show that this compound induces very complex effects in the HL60 cell line. A summary figure perfectly sums up all the results of the manuscript.
However, I would like to make several points.
- The entire study is based on experiments carried out on the HL60 cell line. A few results from a second cell line would have reinforced the message.
- A double annexinV-PI labelling after treatment with 5mg/ml of SPS8 for 15 minutes would give indications about the type of cell death (apoptosis/necroptosis) induced by SPS8.
- Experiments presented in FigS2 show that autophagy blockage with addition of 10mg/ml of chloroquine promotes the SPS8-induced apoptosis. It would have been interesting to perform this experiment with a second autophagy inhibitor such as Bafilomycin A1.
- The authors should show the results of autophagy blockade at the level of a few apoptotic and anti-apoptotic proteins expression. 15 and 24 hours of SPS8 treatment should be sufficient to examine the relashionship between autophagic and apoptotic processes.
- Surprisingly, the expression level of LC3-II is very high in untreated HL60 cells. Would it be possible to have culture conditions which would make it possible to decrease this high autophagic basal level which would make it possible to better appreciate the effects of SP8 on this process ?
- The presentation of the figures is perfect except for fig 3C where there is a slight misalignment of the numbers of the kinetics.
Author Response
We are deeply grateful to the editor and reviewers for providing us the opportunity to revise the study. We have performed several experiments and have modified the description in this manuscript according to the comments to improve its quality. All the comments have been answered point-by-point.
Reviewer #1
Acute Promyelocytic Leukemia (APL) are hematological malignancies who need new effective therapeutic options. The work presented here is focused on the mechanism of action of triazole based aminodiol derivatives against APL and particularly on the promising SPS8 derivative. The authors report that sp8 induces the biosynthesis of sterol and cholesterol as well as the cleavage of the cytoprotective molecule STAT3. They also report that SPS8 stimulates both autophagic and apoptotic events.
The article is extremely well written and the figures are of very high quality. The authors also took care to show the original blots. The results are very interesting and show that this compound induces very complex effects in the HL60 cell line. A summary figure perfectly sums up all the results of the manuscript.
- The entire study is based on experiments carried out on the HL60 cell line. A few results from a second cell line would have reinforced the message.
Ans: We greatly appreciate the reviewer for the comments. We have performed several experiments in THP-1 (acute monocytic leukemia, one of the types of AML) and MV-4-11 (biphenotypic B myelomonocytic leukemia, one of the types of AML) cell lines because we do not obtain another APL cell line except for HL-60 cell line. The data showed that SPS8 also induced cytotoxic effect in THP-1 and MV-4-11 cell lines using MTT assay, flow cytometric analysis of propidium iodide staining and mitochondrial membrane detection. The data have been included into the Supplementary Figure S1 and S2 (p5 and p6).
- A double annexinV-PI labelling after treatment with 5mM of SPS8 for 15 minutes would give indications about the type of cell death (apoptosis/necroptosis) induced by SPS8.
Ans: According to the comment, annexin-V/PI double staining was used to examine the necroptosis effect. The data showed that SPS8 did not induce necroptosis. The data have been included into the Supplementary Figure S2A (p6, the first paragraph).
- Experiments presented show that autophagy blockage with addition of 10 mM of chloroquine promotes the SPS8-induced apoptosis. It would have been interesting to perform this experiment with a second autophagy inhibitor such as Bafilomycin A1.
Ans: We have examined the effect of bafilomycin A1 on SPS8-induced effect. The data showed that bafilomycin A1 at 10, 30 and 50 nM increased the apoptosis by 13.8%, 19.9% and 20.4%, respectively, in HL-60 cells. The potentiation effect of cytotoxicity was similar to that of chloroquine. However, bafilomycin A1, by itself, caused about 10% toxic effect.
- The authors should show the results of autophagy blockade at the level of a few apoptotic and anti-apoptotic proteins expression. 15 and 24 hours of SPS8 treatment should be sufficient to examine the relationship between autophagic and apoptotic processes.
Ans: More apoptosis-related data have been included into the revised manuscript. The data demonstrated that the presence of chloroquine also potentiated SPS8-induced activation of caspase-9 and -3, and g-H2A.X formation (Supplementary Figure S3B) (p8, the second paragraph).
- Surprisingly, the expression level of LC3-II is very high in untreated HL60 cells. Would it be possible to have culture conditions which would make it possible to decrease this high autophagic basal level which would make it possible to better appreciate the effects of SP8 on this process?
Ans: The figure has been improved according to the comment (Figure 3C of the revised manuscript).
- The presentation of the figures is perfect except for fig 3C where there is a slight misalignment of the numbers of the kinetics.
Ans: The alignment of the numbers has been adjusted.

Reviewer 2 Report
The manuscript «Mechanism study of triazole based aminodiol derivatives against acute promyelocytic leukemia – crosstalk between mitochondrial stress-involved apoptosis and autophagy» by Chan and coworkers aims to reveal the mechanisms whereby the compound SPS8 kills HL-60 cells.
Although the authors convincingly show that the killing of the HL-60 cells involves apoptosis, there are several problems with the manuscript.
- The title of the manuscript indicates effects on acute promyelocytic leukemia (APL), while in fact only effects on the APL cell line HL-60 cells are presented. The authors should include key experiments on primary APL cells, and at least include other APL cell lines.
- In the introduction section, the authors present a variety of compounds used to treat APL, along with detailed information on various mechanisms involved in the action of these compounds. Together with a somewhat unstructured discussion on the interplay between autophagy and apoptosis, the introduction becomes inaccessible and difficult to read. This is for instance true for the section including lines 51-68. However, despite all the information presented in the Introduction section, the rationale for testing SPS8 in HL-60 is not apparent.
- In figure 1, the effect of SPS8 on the viability of HL-60 cells is presented. According to the text and panel B, PBMCs are used for comparison of toxicity. However, PBMC are neither mentioned in the figure legends nor in the Materials and methods section. One might also question how relevant it is to compare a cell-line as HL-60 with PBMCs. In line with the issue raised in #1, comparisons between PBMCs and primary APL cells should have been included in figure 1.
- In figure 3, the authors assess the effect of SPS8 on autophagy. In panel A it is not clear that the staining of LC3 demonstrates autophagy. It is commonly agreed that upon autophagy, the LC3-staining pattern changes from diffuse to punctate in line with LC3 being associated with autophagosomes.
- In general, the manuscript lacks hypothesis and appears more as a fishing trip in terms of possible mechanisms to test. It is for instance not clear why the effects of SPS8 on calcium is tested (figure 4). Rather, the authors should have concentrated on one or a few hypotheses, and tested them in a logical manner. If the authors believe in calcium as a mediator, then chelating calcium should prevent the effects of SPS8.
- The gene expression data presented in Table 1 and figure 5 culminate in panel A of figure 6. From the method section and the text in general, it is not clear how the authors end up by focusing on STAT3. However, the effects of SPS8 on STAT3 cleavage are interesting and should have been followed by more experiments to prove its role. STAT3 is not even mentioned in the summary presented in figure 7.
General comments:
- The font sizes on the text in several figures are too small – like in figure 1B, 2A, 6A.
- The English grammar and spelling needs major improvement. This is already evident in the title of the manuscript. It should read: “Mechanistic Study….
Author Response
We are deeply grateful to the editor and reviewers for providing us the opportunity to revise the study. We have performed several experiments and have modified the description in this manuscript according to the comments to improve its quality. All the comments have been answered point-by-point.
Reviewer #2
The manuscript “Mechanism study of triazole based aminodiol derivatives against acute promyelocytic leukemia – crosstalk between mitochondrial stress-involved apoptosis and autophagy” by Chan and coworkers aims to reveal the mechanisms whereby the compound SPS8 kills HL-60 cells.
Although the authors convincingly show that the killing of the HL-60 cells involves apoptosis, there are several problems with the manuscript.
- The title of the manuscript indicates effects on acute promyelocytic leukemia (APL), while in fact only effects on the APL cell line HL-60 cells are presented. The authors should include key experiments on primary APL cells, and at least include other APL cell lines.
Ans: We greatly appreciate the reviewer for the comments. We have performed several experiments in THP-1 (acute monocytic leukemia, one of the types of AML) and MV-4-11 (biphenotypic myelomonocytic leukemia, one of the types of AML) because we do not obtain another APL cell line except for HL-60 cell line. The data showed that SPS8 also induced cytotoxic effect in THP-1 and MV-4-11 cell lines using MTT assay, flow cytometric analysis of propidium iodide staining and mitochondrial membrane detection. The data have been included into the Supplementary Figure S1 and S2 (p5 and p6). Besides, the “acute promyelocytic leukemia” in the title has be replaced with “leukemic cells”. The related description in the text has been modified accordingly.
- In the introduction section, the authors present a variety of compounds used to treat APL, along with detailed information on various mechanisms involved in the action of these compounds. Together with a somewhat unstructured discussion on the interplay between autophagy and apoptosis, the introduction becomes inaccessible and difficult to read. This is for instance true for the section including lines 51-68. However, despite all the information presented in the Introduction section, the rationale for testing SPS8 in HL-60 is not apparent.
Ans: The Introduction section has been re-written and the rationale for testing SPS8 has been addressed in the revised manuscript (p3 and p4).
- In figure 1, the effect of SPS8 on the viability of HL-60 cells is presented. According to the text and panel B, PBMCs are used for comparison of toxicity. However, PBMC are neither mentioned in the figure legends nor in the Materials and methods section. One might also question how relevant it is to compare a cell-line as HL-60 with PBMCs. In line with the issue raised in #1, comparisons between PBMCs and primary APL cells should have been included in figure 1.
Ans: Human PBMCs were purchased from Zen-Bio Inc. (NC, USA). The description has been included into the Materials and Methods section (p15, the 6th line) and the figure 1B legend. Because it is difficult for us to obtain primary APL cells, we have performed additional experiments in THP-1 and MV-4-11 cell lines. The data showed that SPS8 also induced cytotoxic effect in THP-1 and MV-4-11 cell lines using MTT assay, flow cytometric analysis of propidium iodide staining and mitochondrial membrane detection. The data have been included into the Supplementary Figure S1 and S2 (p5 and p6).
- In figure 3, the authors assess the effect of SPS8 on autophagy. In panel A it is not clear that the staining of LC3 demonstrates autophagy. It is commonly agreed that upon autophagy, the LC3-staining pattern changes from diffuse to punctate in line with LC3 being associated with autophagosomes.
Ans: The quality of the images in Figure 3A has been improved that the punctate dots of LC3 staining have been apparent in the revised manuscript.
- In general, the manuscript lacks hypothesis and appears more as a fishing trip in terms of possible mechanisms to test. It is for instance not clear why the effects of SPS8 on calcium is tested (figure 4). Rather, the authors should have concentrated on one or a few hypotheses, and tested them in a logical manner. If the authors believe in calcium as a mediator, then chelating calcium should prevent the effects of SPS8.
Ans: We appreciate the reviewer’s comments. We have examined the effect of BAPTA, an intracellular calcium chelator, on SPS8-induced cytotoxicity using MTT assay. The data showed that BAPTA significantly prevented the cells from SPS8-induced cytotoxic effect although BAPTA alone induced a moderate toxicity. The data have been included into the revised manuscript (Figure 4B and p8, the last two lines).
- The gene expression data presented in Table 1 and figure 5 culminate in panel A of figure 6. From the method section and the text in general, it is not clear how the authors end up by focusing on STAT3. However, the effects of SPS8 on STAT3 cleavage are interesting and should have been followed by more experiments to prove its role. STAT3 is not even mentioned in the summary presented in figure 7.
Ans: We appreciate the reviewer’s comments. In addition to STAT3 cleavage, we have performed additional examination on the phosphorylation of STAT3. The data have been included into the Figure 6B showing that SPS8 inhibited the phosphorylation of STAT3 and induced a profound cleavage of STAT3. The data indicated the decline of STAT3 activity to SPS8 action (p10, the first line). Besides, the STAT3 has been included into the graphic Figure 7.
- The font sizes on the text in several figures are too small – like in figure 1B, 2A, 6A.
Ans: The font sizes on the text in the figures have been improved in all the figures.
- The English grammar and spelling needs major improvement. This is already evident in the title of the manuscript. It should read: “Mechanistic Study….
Ans: The title has been changed according to the comment and the revised manuscript has been edited through the English Editing Service of MDPI Journal.

Round 2
Reviewer 1 Report
The authors presented a new version of their manuscript in which they took into account all the comments.
A number of experiments have been carried out to reinforce the message of the manuscript. The quality of unpublished blots and additional figures is of high quality.
The authors made a real effort in the quality of their responses and the manuscript was thus really improved with very robust conclusions.
I no longer see any obstacle to the publication of this work.
Reviewer 2 Report
The revised manuscript is extensively improved according to my previous suggestions. I therefore suggest that the manuscript is acceptable for publication.